# Pbp1 associates with Puf3 and promotes translation of its target mRNAs involved in mitochondrial biogenesis

Floortje van de Poll[1], Benjamin M. Sutter[1], Michelle Grace Acoba[1], Daniel Caballero[1], Samira Jahangiri[1], Yu-San Yang[1], Chien-Der Lee[1], Benjamin P. Tu[1,2]*

1 Department of Biochemistry, University of Texas Southwestern Medical Center, Dallas, Texas, United States of America, 2 Howard Hughes Medical Institute, University of Texas Southwestern Medical Center, Dallas, Texas, United States of America

☺ These authors contributed equally to this work.
* benjamin.tu@utsouthwestern.edu

**Data Availability Statement:** Raw sequencing data have been deposited at Gene Expression Omnibus with accession number: GSE227356

## Abstract

Pbp1 (poly(A)-binding protein—binding protein 1) is a cytoplasmic stress granule marker that is capable of forming condensates that function in the negative regulation of TORC1 signaling under respiratory conditions. Polyglutamine expansions in its mammalian ortholog ataxin-2 lead to spinocerebellar dysfunction due to toxic protein aggregation. Here, we show that loss of Pbp1 in *S. cerevisiae* leads to decreased amounts of mRNAs and mitochondrial proteins which are targets of Puf3, a member of the PUF (Pumilio and FBF) family of RNA-binding proteins. We found that Pbp1 supports the translation of Puf3-target mRNAs in respiratory conditions, such as those involved in the assembly of cytochrome c oxidase and subunits of mitochondrial ribosomes. We further show that Pbp1 and Puf3 interact through their respective low complexity domains, which is required for Puf3-target mRNA translation. Our findings reveal a key role for Pbp1-containing assemblies in enabling the translation of mRNAs critical for mitochondrial biogenesis and respiration. They may further explain prior associations of Pbp1/ataxin-2 with RNA, stress granule biology, mitochondrial function, and neuronal health.

## Author summary

The yeast protein Pbp1 and its mammalian ortholog ataxin-2 are both markers of stress granules. The function of such granules and what they do to mRNA transcripts has been unclear. Here we use budding yeast to show that Pbp1 regulates transcripts involved in mitochondrial biogenesis, specifically those that are targets of Puf3, a member of the PUF family of sequence-specific RNA-binding proteins. Loss of Pbp1 results in reduced amounts of these mRNAs and the proteins they encode. Intriguingly, many of these are components of cytochrome c oxidase. We further show Pbp1 associates with Puf3 under respiratory conditions to assist the translation of these mRNAs. These findings suggest

**Funding:** This work was supported by the Howard Hughes Medical Institute (HHMI) to BPT; the HHS | NIH | National Institute of Neurological Disorders and Stroke (NINDS): R01NS115546 to BPT and the HHS | NIH | National Institute of Neurological Disorders and Stroke (NINDS): 3R01NS115546-04S1 to DC. The funders had no role in study design, data collection and analysis, decision to publish, or preparation of the manuscript.

**Competing interests:** The authors declare that they have no conflict of interest.

Pbp1-containing granules regulate the translation of a specific class of mRNAs according to the metabolic demands of the cell.

## Introduction

Yeast cells are capable of rapidly adapting their metabolism to changes in environmental conditions. When grown in glucose media, yeast cells use glycolysis for energy production and suppress mitochondrial biogenesis. However, in the presence of a non-fermentable carbon source, such as lactate, yeast cells adapt by inducing mitochondrial biogenesis to increase ATP production by oxidative phosphorylation (OXPHOS). The majority of the protein components of the electron transport chain are nuclear-encoded and need to be imported into the mitochondria. This essential process also requires cytosolic protein participants and is tightly regulated according to the cell's metabolic needs [1].

Puf3 is one such cytosolic protein that plays a key role in the post-transcriptional regulation of mitochondrial biogenesis. Puf3 is a member of the PUF protein family, which are conserved RNA-binding proteins that recognize a consensus motif in the 3' UTR of their target mRNAs [2]. Most of the target mRNAs associated with Puf3 are important for OXPHOS, many of which encode for subunits of mitochondrial ribosomes or respiratory complexes [3–8]. Depending on glucose availability, Puf3 can switch the fate of its target mRNAs from decay to translation [9]. In the presence of non-fermentable sugars that require respiratory metabolism, Puf3 becomes heavily phosphorylated in its N-terminal low complexity region [9,10], leading to the stabilization and translation of its bound transcripts. Furthermore, Puf3 has been found in ribonucleoprotein (RNP) granules [9].

The yeast ortholog of ataxin-2, poly(A) binding protein-binding protein 1 (Pbp1), associates with poly(A)-tails and is also found in RNP granules [11,12]. Prior evidence also suggests that Pbp1 is important for the maintenance of mitochondrial function. For instance, Pbp1 overexpression has been shown to rescue various mitochondrial abnormalities including defects in intron splicing, protein import, and mitochondrial damage and death caused by a mutation in the mitochondrial ADP/ATP carrier *AAC2* that is equivalent to one in humans with progressive external ophthalmoplegia [13–15]. We also demonstrated that Pbp1 negatively regulates TORC1 leading to autophagy and mitophagy in nutritional stress conditions that require respiration [16]. In addition, loss of Pbp1 can lead to mitochondrial dysfunction, increased sensitivity to oxidative stress, and cell death [16,17]. These findings collectively suggest that Pbp1 plays an important role in promoting mitochondrial function.

Here we utilized *S. cerevisiae* to discover a functional relationship between Pbp1, mitochondria, and Puf3. We find that Pbp1 and Puf3 interact through their low-complexity domains and that this interaction facilitates the translation of mRNAs that are targets of Puf3 under respiratory conditions. These findings reveal that Pbp1 supports mitochondrial function via assemblies involved in mRNA translation, which may provide key insights into how ataxin-2 mutations cause neurodegenerative diseases.

## Results

To interrogate a possible role for Pbp1 in mitochondrial function, we assayed mitochondrial protein abundance in response to switching from glycolytic (YPD) to respiratory (YPL) media, which are conditions that demand mitochondrial biogenesis, in wild type (WT) and *pbp1Δ* cells. We assessed Por1 (mitochondrial porin) and Cox2 (subunit II of cytochrome c oxidase) protein levels using readily available antibodies by immunoblot in YPD and at several time

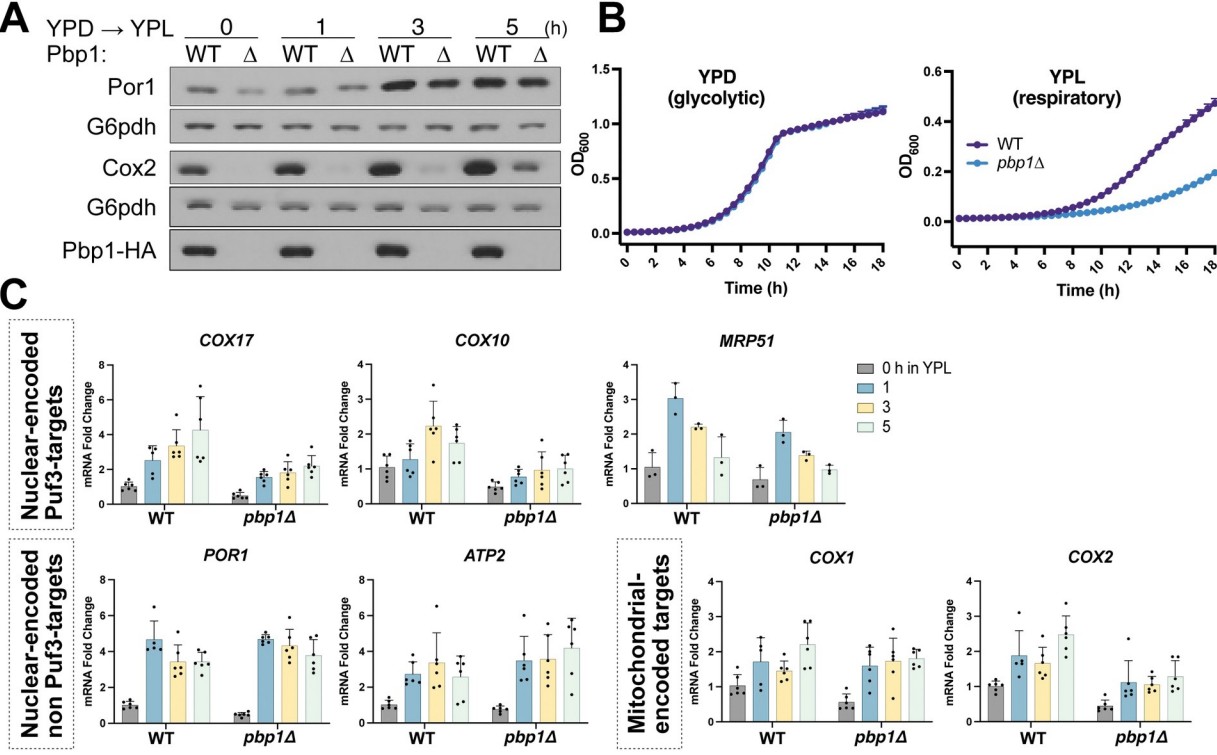

**Fig 1. Pbp1 regulates Puf3-target mRNA and protein levels.** A. A protein (Cox2) whose synthesis is dependent on Puf3-target mRNAs is decreased in *pbp1Δ* cells compared to WT. Cells were grown to log phase in glucose (YPD) media and then washed and resuspended in lactate (YPL) media. Samples were collected before and after switch to YPL, quenched, followed by protein extraction. Equal amounts of total protein were assessed by immunoblot for Cox2, Por1, Pbp1, and G6pdh levels. B. Cells lacking Pbp1 have a growth defect in respiratory (YPL) but not in glycolytic (YPD) growth conditions compared to WT cells. Cells were grown to log phase in glucose or lactate media, then resuspended at low OD in fresh media and $A_{600}$ was measured every 30 min for 18 h. Error bars represent SD; n = 3. C. Decreased mRNA levels of Puf3-target mRNAs are observed in *pbp1Δ* cells. The abundance of the indicated transcripts was measured by qRT-PCR and normalized to *ACT1* levels at different time points after switching from YPD to YPL media. Error bars represent SD; n = 6 except for *MRP51*; n = 3.

points after switching to YPL (Fig 1A). Por1 protein levels increased over time during growth in respiratory conditions in both WT and *pbp1Δ* cells. Strikingly, Cox2 protein levels were severely decreased in *pbp1Δ* compared to WT cells at all time points. Moreover, *pbp1Δ* cells showed a significantly reduced growth rate in YPL compared to WT cells (Fig 1B), but their growth rate in YPD was similar to WT. These observations are consistent with *pbp1Δ* cells having compromised mitochondrial function due to reduced amounts of proteins required for respiration, such as Cox2.

As Pbp1 interacts with Pab1 (poly(A)-binding protein 1) and contains putative RNA-binding domains [11], we tested the possibility that the loss of Pbp1 might affect the abundance of mRNAs involved in mitochondrial function. Analysis of a panel of nuclear- and mitochondrial-encoded mRNAs revealed that *COX17*, *COX10*, and *MRP51* transcripts were decreased in *pbp1Δ* cells (Fig 1C). By contrast, no significant differences were observed in *POR1* and *ATP2* transcripts. Notably, the transcripts exhibiting reduced abundance in *pbp1Δ* mutant cells (*COX17*, *COX10*, *MRP51*) all have Puf3-binding elements in their 3' UTRs and are Puf3-target mRNAs [6,9]. For mitochondrial-encoded transcripts, we observed a modest decrease in COX2 transcripts in *pbp1Δ* cells at later time points in YPL and slight differences in COX1. The translation of COX2 relies on mitochondrial ribosomes, most subunits of which are post-transcriptional mRNA targets of Puf3 (e.g., MRP51). Therefore, defects in the translation of

mitochondrial-encoded genes, such as *COX2*, indicate disruption of Puf3 function [9]. In principle, reduction of Cox2 protein levels could also be due to altered mitochondrial membrane potential and reduced protein import into this organelle. As a problematic mitochondrial protein import system leads to the cytosolic accumulation of mitochondria-targeted proteins [14,18], we examined the fate of Cox4, a nuclear-encoded complex IV subunit that does not contain a Puf3-binding element in the 3'UTR of its encoding gene. Steady state Cox4 protein amounts were comparable between WT and *pbp1Δ* cells (S1 Fig). Moreover, following treatment of cells with the uncoupler CCCP, we did not observe an increase in the "pre"-processed form of Cox4 in *pbp1Δ* cells (S1 Fig), which suggests there is no major defect in mitochondrial membrane potential and protein import in these mutants.

We next performed an unbiased, transcriptome-wide analysis to assess the abundance of mRNAs in *pbp1Δ* cells growing in either glycolytic (YPD) or respiratory conditions (YPL) (Fig 2). Gene ontology analysis of the set of transcripts with a significant decrease in *pbp1Δ* cells revealed that mitochondrial translation and mitochondrial gene expression categories were highly enriched (Fig 2C and 2D). Moreover, following examination of transcripts encoding mitochondrial ribosomal subunits, all of which are Puf3 targets [8], or a set of Puf3 targets determined by multi-omics analyses [6], these mRNAs were all found to be decreased in *pbp1Δ* cells, especially in respiratory conditions (Fig 2A and 2B).

The specific decrease in mRNA levels of Puf3-target mRNAs in *pbp1Δ* cells suggests that Pbp1 may help Puf3 stabilize and promote the translation of its target mRNAs. To test this possibility, we examined the extent of translation of a Puf3-binding element-containing mRNA following the shutoff of its transcription. The reporter consists of a mitochondrial targeting sequence (MTS), the GFP coding sequence, and the 3' UTR of the mitochondrial ribosomal gene MRP51, which contains a Puf3-binding element (P3E) (Fig 3A) [9]. The construct was integrated into a strain expressing a chimeric GAL4$_{DBD}$-ER-VP16 transcription factor enabling inducible expression by the addition of ß-estradiol [19]. This reporter allows monitoring of GFP transcript and protein amounts following a pulse of expression in the background of WT or *pbp1Δ* cells. Moreover, including a mutant reporter in which the P3E contains a four-nucleotide mutation allows the assessment of reporter translation dependent on an intact P3E.

WT and *pbp1Δ* cells were grown in YPD media, and β-estradiol was added for 45 min before washout and then switched to YPL respiratory media. We observed an expected, gradual decrease in mRNA due to the absence of any inducer (Fig 3B). Strikingly, levels of the reporter mRNA were significantly lower in *pbp1Δ* cells, suggesting that Pbp1 may be stabilizing Puf3-target mRNAs. Interestingly, mRNA expression levels of the mutant P3E reporter were similar in WT and *pbp1Δ* cells, indicating that the reduced reporter expression in *pbp1Δ* cells is mediated by Puf3 and an intact P3E sequence in its target mRNAs.

Upon examination of translation of the reporter mRNA containing an intact P3E, WT cells synthesized significant amounts of GFP protein as a function of time in YPL media, despite decreasing reporter mRNA levels. In contrast, *pbp1Δ* mutant cells synthesized significantly lower amounts of GFP protein than WT (Fig 3C). No significant differences in translation of the reporter with a mutant P3E were observed. Interestingly, a basal amount of reporter translation was still observed in *pbp1Δ* cells. It has been suggested that Tom20, a component of the outer mitochondrial membrane translocase complex, can recruit Puf3-target mRNAs to the mitochondrial surface even in the absence of Puf3 [20]. Taken together, these data demonstrate that Pbp1 stabilizes Puf3-target mRNAs and promotes their translation into protein.

To further assess whether Pbp1 acts to stabilize Puf3 target mRNAs, we performed a transcriptional shutoff assay using the antibiotic thiolutin in cells growing at log phase in lactate media. We hypothesized that loss of Pbp1 might enhance the decay of Puf3 target mRNAs, but

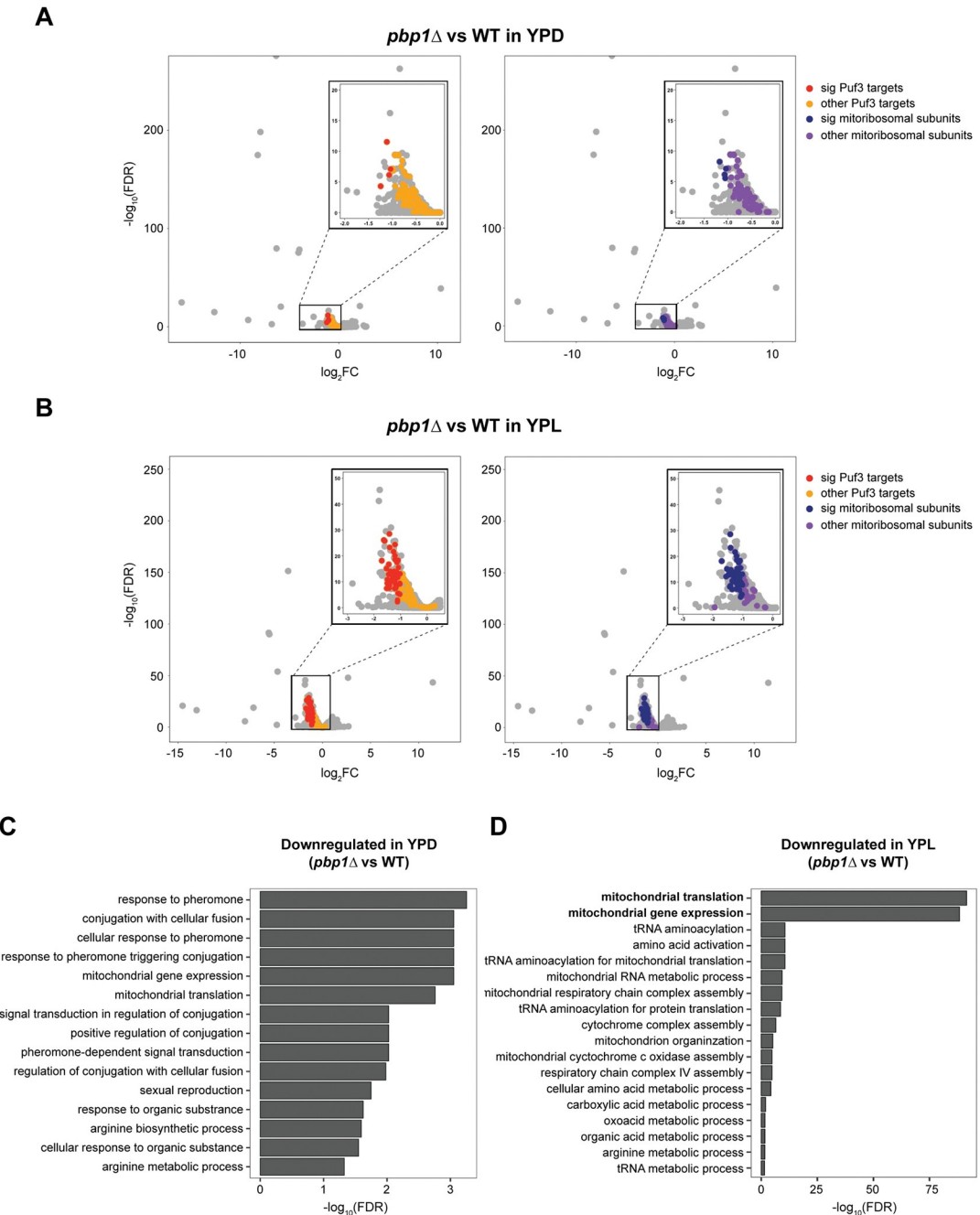

**Fig 2. Transcriptomic analysis of *pbp1Δ* mutant cells.** A. RNA-seq analysis of *pbp1Δ* and WT cells grown in YPD to log phase. Volcano plots depict differentially expressed genes. "Sig" denotes whether the change in expression is statistically significant. B. RNA-seq analysis of *pbp1Δ* and WT cells following switch to YPL for 3 h. Volcano plots depict differentially expressed genes. Complete RNA-seq data are also available and searchable in S1 Table. C. Gene ontology (GO) terms that were significantly enriched in significantly downregulated genes in *pbp1Δ* mutant cells in the YPD condition. D. Gene ontology (GO) terms that were significantly enriched in significantly downregulated genes in *pbp1Δ* mutant cells in the YPL condition. Note the abundance of transcripts involved in some aspect of mitochondrial function that are decreased in *pbp1Δ* cells.

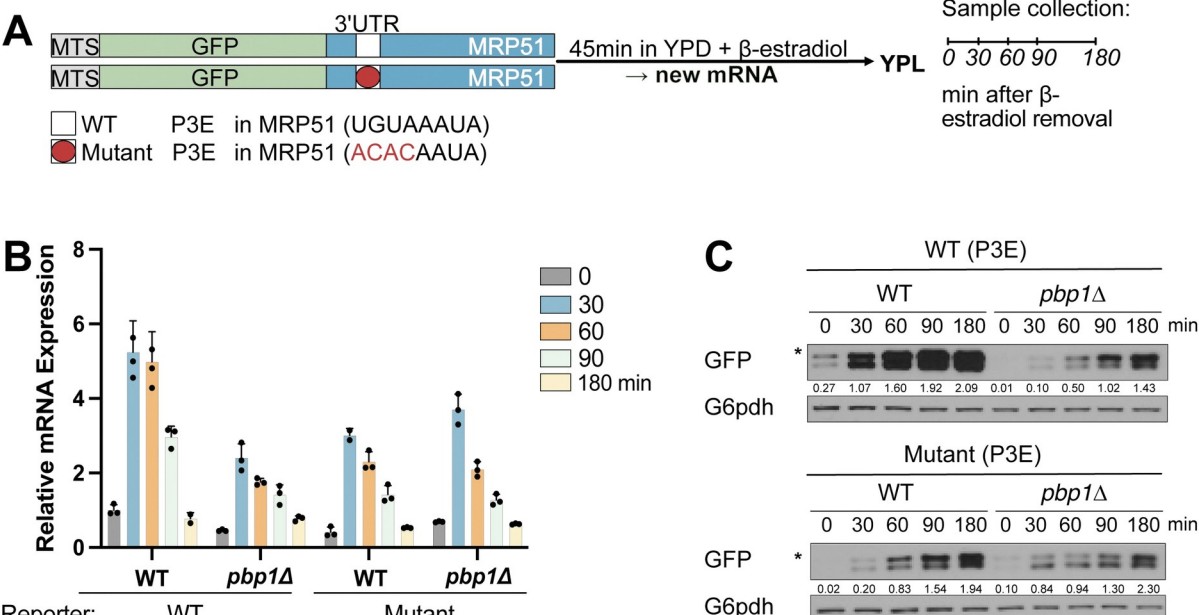

**Fig 3. Pbp1 promotes the translation of Puf3-target mRNAs in respiratory conditions.** A. Schematic representation of the Puf3 mRNA reporter assay. The indicated reporter constructs containing either a WT or mutant 3' UTR of the MRP51 gene were placed under the control of the GAL1 promoter and integrated into strains expressing GAL4DBD-ER-VP16, in a WT or *pbp1Δ* background. After cells were grown in YPD to log phase, β-estradiol was added to induce a pulse of transcription of the reporter mRNA. After 45 min, cells were washed twice to remove β-estradiol and then resuspended in YPL medium. P3E = Puf3-binding Element, MTS = mitochondrial targeting sequence. B. A reduction in GFP-mRNA can be observed for the reporter with intact P3E in *pbp1Δ* cells compared to WT, while no significant differences are observed for the mutant P3E reporter. Cells were harvested at 0, 30, 60, 90, and 180 min for analysis of reporter mRNA levels by qRT-PCR for GFP and normalized to *ACT1*. The increase in mRNA levels observed after medium switch is due to an unavoidable washing step to remove β-estradiol. Note that Puf3 is not absolutely required for the translation of its target mRNAs. Error bars represent SD; n = 3. C. Pbp1 promotes the translation of the reporter containing an intact P3E. Reporter translation was assayed using an anti-GFP antibody, and G6pdh was used as a loading control. * denotes the slightly larger, unprocessed form of GFP containing a MTS.

surprisingly we found that Puf3 target mRNAs were relatively decay-resistant in *pbp1Δ* cells while non-Puf3 target mRNAs exhibited decay kinetics that were more similar between WT and *pbp1Δ* cells (S2 Fig). These results are not consistent with a straightforward function for Pbp1 in stabilizing Puf3 target mRNAs, but suggest that Pbp1 may be involved in the turnover of these mRNAs during respiratory growth. The absence of Pbp1 could also elicit compensatory effects on the stability of these transcripts.

We next tested whether there might be a genetic interaction between Pbp1 and Puf3. We generated *puf3Δ*, *pbp1Δ*, and *pbp1Δpuf3Δ* cells and examined mitochondrial protein levels by immunoblot following growth in YPD and YPL media (Fig 4A). As expected, Cox2, Por1, and Atp2 (subunit of the mitochondrial ATP synthase) protein levels all increased upon switching to respiratory conditions. However, Cox2 protein levels, which are dependent on the translation of Puf3-target mRNAs, decreased in *pbp1Δ*, *puf3Δ*, and *pbp1Δpuf3Δ* cells compared to WT cells. Por1 and Atp2 protein levels, which do not depend on Puf3-targets, were similar amongst the tested strains. Interestingly, the double knockout *pbp1Δpuf3Δ* exhibited a similar reduction in Cox2 as *puf3Δ* cells, suggesting that Puf3 is epistatic to Pbp1. In addition, we confirmed that Pbp1 levels were not affected by the absence of Puf3, whereas Puf3 levels were only modestly decreased in the absence of Pbp1 but only in lactate media (S3 Fig). Taken together, these data suggest that Pbp1 and Puf3 function together in the same pathway. Curiously, *pbp1Δ* cells exhibit lower levels of Cox2 than *pbp1Δpuf3Δ* double mutant cells, suggesting that in the absence of Pbp1, Puf3 may direct some of its target mRNAs towards decay pathways

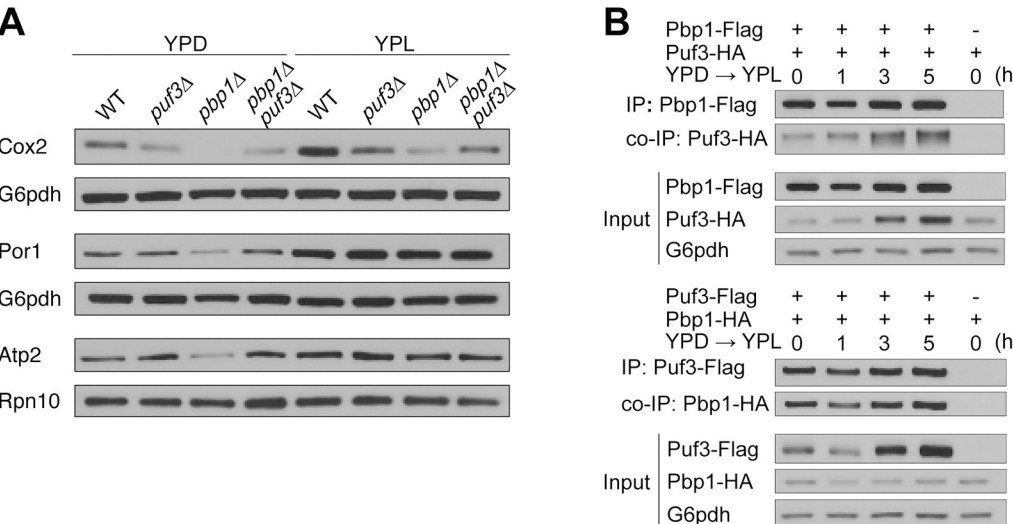

**Fig 4. Increased amounts of Puf3 interact with Pbp1 under respiratory conditions.** A. Only proteins whose synthesis is dependent on Puf3-target mRNAs (Cox2) are decreased in *puf3Δ*, *pbp1Δ*, and *pbp1Δpuf3Δ* cells as compared to WT. Cells were grown to log phase in YPD, then washed and resuspended in YPL. Samples were collected before (0 h) and after switch to YPL (3 h), quenched, followed by protein extraction. Equal amounts of protein were assayed by immunoblot for Cox2, Por1, Atp2, Rpn10 and G6pdh levels. How transcript levels respond in the double mutant is shown in S4 Fig. B. Increased amounts of Puf3 are associated with Pbp1 in respiratory conditions. Cells with epitope-tagged Pbp1 and Puf3 were grown in YPD and then washed and resuspended in YPL. Samples were collected at the indicated time points, after which Flag immunoprecipitation was performed.–denotes negative control cells lacking a Flag-tag.

(S4 Fig). Notably, a decrease in each of the surveyed mitochondrial proteins can also be observed in *pbp1Δ* cells in YPD, which is suggestive of a general impairment in mitochondria even in glucose conditions. However, for this study, we focus on the role of Pbp1 in respiratory conditions due to the exacerbated phenotype of reduced Cox2 levels and Puf3-target mRNAs specifically observed in YPL media.

Having observed genetic interactions and phenotypic consequences on Puf3-targets in *pbp1Δ* mutant cells, we next investigated whether Pbp1 and Puf3 interact by co-immunoprecipitation analysis using C-terminal, epitope-tagged versions of Pbp1 (Pbp1-FLAG) and Puf3 (Puf3-HA) (Fig 4B). Pbp1-Flag co-IP experiments revealed that Pbp1 was indeed able to pull down Puf3. In addition, we performed the reciprocal Flag co-IP and obtained corresponding results, demonstrating that Pbp1 and Puf3 interact. We observed that Puf3 protein levels increase over time in response to the switch to respiratory conditions (Figs 4B and S3). Interestingly, we observed an increased interaction between Pbp1 and Puf3 over time in YPL media (Fig 3B), which can be attributed to increased amounts of the Puf3 protein. This suggests that the Pbp1-Puf3 interaction is responsive to the cellular metabolic state.

To determine which region within Pbp1 is required for binding Puf3, we engineered a series of Pbp1 truncations (Fig 5A). Using co-immunoprecipitation, we observed that deletion of the mid-section (aa 299–570) or the low-complexity domain (aa 570–722) largely abolished the ability of Pbp1 to pull down Puf3 (Fig 5B). We then asked if the Pbp1-Puf3 interaction facilitates the translation of Puf3-target mRNAs by assessing Cox2 and Por1 protein levels in these mutants. We observed reduced Cox2, but not Por1, protein levels in Pbp1 *midΔ* and Pbp1 *LCDΔ* cells, while no differences were observed in cells expressing the other Pbp1 truncation constructs (Fig 5C). Reduced Cox2 amounts correlated with reduced growth rate of Pbp1 *midΔ* and Pbp1 *LCDΔ* in respiratory conditions (Fig 5D). Consistent with these effects on

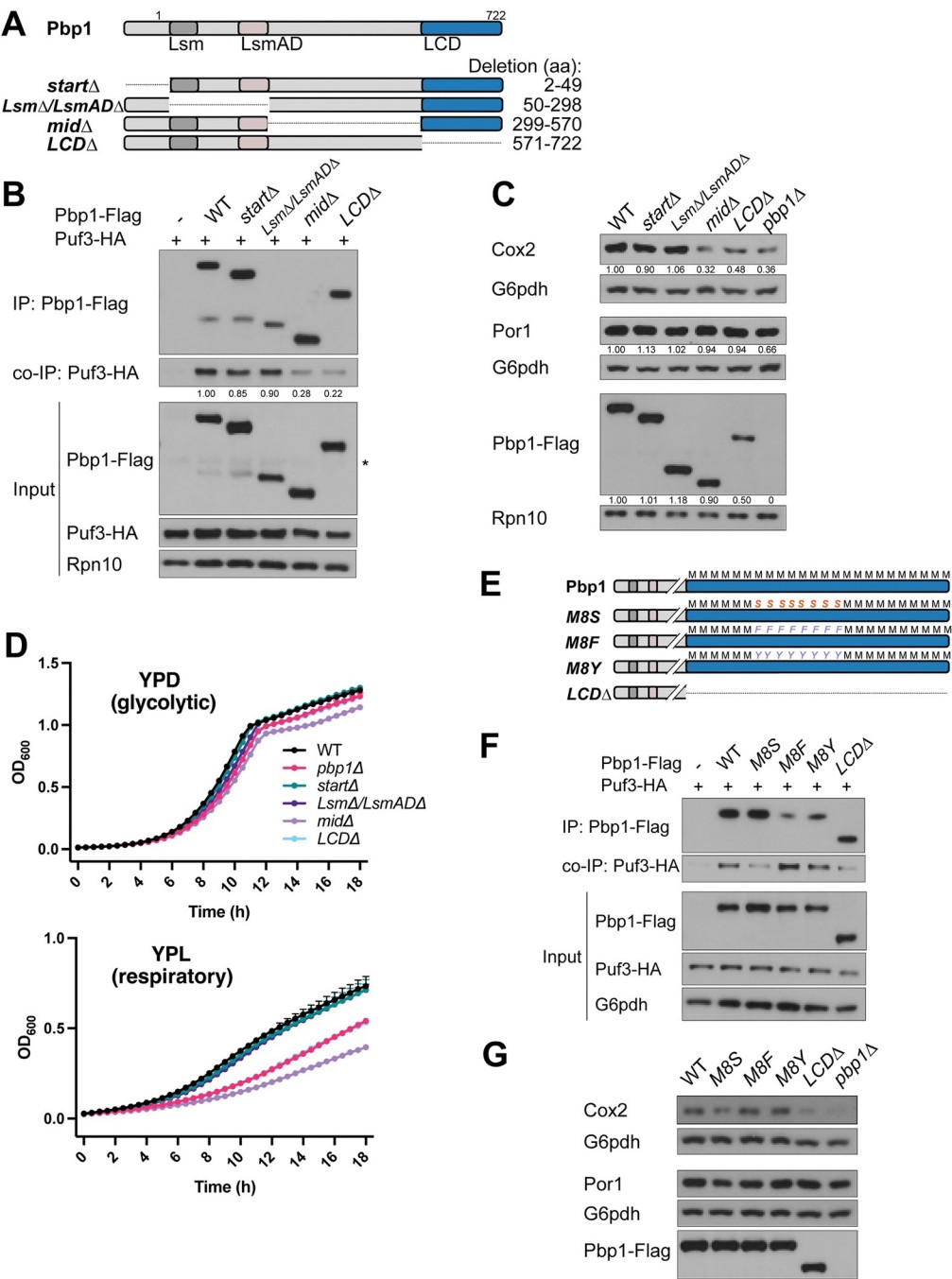

**Fig 5. C-terminal, low complexity regions of Pbp1 mediate its interaction with Puf3.** A. Schematic representation of Pbp1 and the deletion mutants used. B. Interaction between Puf3 and Pbp1 lacking its mid-section (*midΔ*) or C-terminal LCD (*LCDΔ*) is decreased in respiratory conditions. Cells with epitope-tagged Pbp1 and Puf3 were grown in YPD and then washed and resuspended in YPL. Samples were collected at the indicated time points, after which Flag immunoprecipitation was performed.–denotes negative control lacking a Flag-tag, but including HA-tag. * denotes non-specific band. C. Cox2 protein levels are decreased in Pbp1 *midΔ*, *LCDΔ*, and *pbp1Δ* in respiratory conditions. Cells were grown to log phase in YPD, then washed and resuspended in YPL. Samples were collected after 3 h in YPL, quenched, followed by protein extraction. Equal amounts of protein were assessed by immunoblot for Cox2, Por1, Atp2, Pbp1, Rpn10, and G6pdh levels. Transcript levels are shown in S5 Fig. D. Growth curves of the indicated strains in YPD or YPL media. E. Schematic representation of Pbp1 LCD-mutants used. Methionine residues within this key region of the LCD (M591, M595, M605, M606, M614, M616, M618, M625) were mutated to serine (M8S), phenylalanine (M8F), or tyrosine (M8Y). Pbp1 M8S forms weaker assemblies than WT, whereas M8F and M8Y both form stronger assemblies. F. Pbp1-Puf3 interaction is decreased in the M8S mutant. Cells expressing the indicated,

epitope-tagged variants of Pbp1 and Puf3 were grown in YPD and then washed and resuspended in YPL. Samples were collected at 3 h in YPL, after which Flag-immunoprecipitation was performed.–denotes negative control lacking Flag-tag. G. Cox2 protein levels are decreased in M8S, LCD, and *pbp1Δ* mutants. Cells expressing the indicated epitope-tagged variants of Pbp1 were grown to log phase in YPD, then washed and resuspended in YPL. After 3 h, cells were harvested and quenched, followed by protein extraction. Equal amounts of protein were assessed by immunoblot for Cox2, Por1, Pbp1, and G6pdh levels.

protein levels, qRT-PCR analysis showed reduced mRNA levels of Puf3-target transcripts *COX17* and *COX10*, but not POR1 or ATP2, in the corresponding truncation mutants (S5 Fig).

We previously found that Pbp1 self-associates through its low complexity domain, which is mediated by key methionine residues in the LCD [16,17]. Replacement of eight methionine residues in the LCD by serine (M8S) leads to weaker self-assembly, increased TORC1 signaling, and reduced autophagy. In contrast, the M8F and M8Y mutants show stronger self-assembly and slightly increased autophagy [16,17]. Therefore, we asked if these key methionine residues in the LCD responsible for the ability of Pbp1 to form condensates might also alter its interaction with Puf3 (Fig 5E and 5F). Interestingly, the M8S mutant showed reduced interaction with Puf3, in line with its compromised ability to phase separate and form assemblies. In contrast, the M8F and M8Y mutants showed increased interaction with Puf3, which is consistent with their ability to form stronger assemblies. Furthermore, Cox2 protein expression in these mutants, but not Por1, was affected in a manner that correlated with their ability to interact with Puf3. Decreased Cox2 protein levels were observed in the M8S mutant (Fig 5G), whereas increased Cox2 protein levels were observed in the M8F and M8Y mutants, compared to WT. These data suggest a strong correlation between the ability of Pbp1 to self-associate, the extent of its interaction with Puf3, and resulting amounts of Cox2 protein.

To further characterize the interaction between Pbp1 and Puf3, we used different truncation mutants of Puf3 (Fig 6A). Puf3 contains a PUF domain that binds to the 3' UTR of mRNAs encoding mitochondrial proteins as well as a low-complexity N-terminal (Nt) region [7,9]. It also contains a stretch of glutamine residues (polyQ) amid the protein. Puf3 *PUFΔ* and Puf3 *NtΔ* proteins were expressed at lower amounts than WT Puf3 or Puf3 *polyQΔ* proteins (Fig 6B and 6C). Deleting the Nt region abolished the ability of Puf3 to pull down Pbp1. The Puf3 *PUFΔ* mutant also showed reduced interaction with Pbp1. In addition, Cox2 and Por1 protein levels were assayed in these mutants (Fig 6C). We observed reduced Cox2 but normal amounts of Por1 protein levels in Puf3 *NtΔ* and Puf3 *PUFΔ*, similar to *puf3Δ* cells, in YPL media. We then assayed mRNA levels by qRT-PCR analysis, and decreased amounts of Puf3-target mRNAs (e.g., *COX10*) were observed for the *NtΔ*, *PUFΔ*, and complete deletion (S5 Fig). We conclude that without the N-terminal low complexity region or PUF domain, Puf3 can no longer interact with Pbp1 in respiratory conditions. Our cumulative findings suggest that Pbp1 and Puf3 interact via their low complexity domains (C-term LCD of Pbp1, N-term LCD of Puf3) to stabilize and promote translation of Puf3-target mRNAs for mitochondrial biogenesis.

Having shown that Pbp1 and Puf3 interact through their respective low-complexity domains, we next tested whether their interaction might be dependent on RNA. Treatment with RNase I$_f$ did not affect the interaction between Pbp1 and Puf3 as assessed by co-immunoprecipitation (Fig 7A). Next, we performed RNA-IP experiments to assess whether Pbp1 could pull down Puf3 target or non-Puf3 target mRNAs. Pbp1 associated with a panel of tested mRNAs. Importantly, for all of the Puf3-target mRNAs tested (*COX17*, *COX10*, *MRP1*, *RSM19*, *MEF1*), the immunoprecipitated amount was reduced in *puf3Δ* cells, especially in respiratory conditions (YPL) (Fig 7B). Taken together, these data suggest Pbp1 may directly

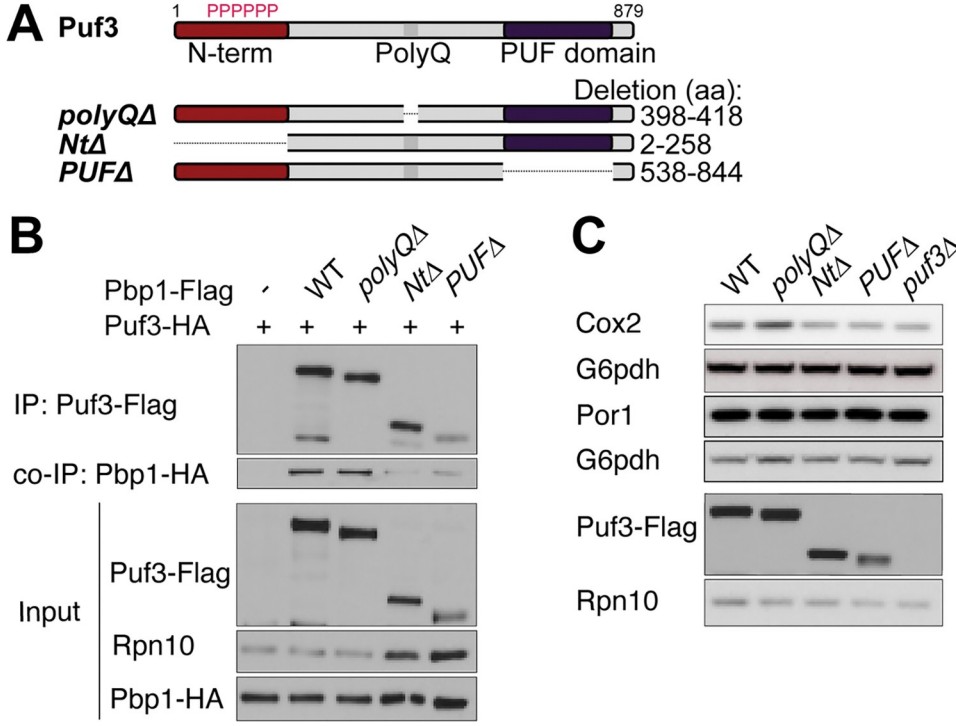

**Fig 6. The N-terminal low complexity and PUF domains of Puf3 are critical for interaction with Pbp1.** A. Schematic representation of Puf3 and deletion mutants used. B. Decreased interaction between Puf3 and Pbp1 is observed when the N-terminal LCD or PUF domain is lacking. Cells with endogenously tagged Pbp1 and Puf3 were grown in YPD and then washed and resuspended in YPL. Samples were collected after 3 h in YPL, after which Flag-immunoprecipitation was performed.–denotes negative control lacking Flag-tag. Note: total protein amounts are doubled for *Puf3 NtΔ* and *Puf3 PUFΔ* samples as compared to the others, as the expression of these two mutants is reduced (see Input). C. Cox2 protein levels are decreased in *Puf3 NtΔ* and *Puf3 PUFΔ* mutants in respiratory conditions. Cells were grown to log phase in YPD and then switched to YPL. Samples were collected after 3 h in YPL, quenched, followed by protein extraction. Equal amounts of protein were assessed by immunoblot for Cox2, Por1, Rpn10, and G6pdh levels. Note that *NtΔ* and *PUFΔ* mutants of Puf3 are unstable and expressed at lower levels compared to WT. Transcript levels are shown in S5 Fig.

interact with Puf3 and that the association of Pbp1 with Puf3-target mRNAs indeed depends on Puf3.

Finally, previous studies have shown that Pbp1 can function as a negative regulator of TORC1 in response to heat or nutritional stress [16,21]. We tested whether the decreased Cox2 protein amounts in *pbp1Δ* cells might be due to increased TORC1 activity by treating cells with rapamycin. No difference in Cox2 levels was observed in *pbp1Δ* cells following rapamycin addition, indicating that the Cox2 protein reduction in *pbp1Δ* cells is independent of TORC1 signaling and instead likely due to its interaction with Puf3 (S6 Fig).

## Discussion

In this study, we show that Pbp1 supports mitochondrial function by promoting the translation of Puf3-target mRNAs that are involved in mitochondrial biogenesis. Both Pbp1 and Puf3 harbor low complexity domains, and their association is required for normal Puf3 function. We speculate that Pbp1 self-associates under respiratory conditions to recruit Puf3 to the vicinity of mitochondria, where Puf3 promotes the translation of mRNAs central for mitochondrial biogenesis. Consistent with this hypothesis, we observed that Pbp1's capacity to self-assemble correlated with its interaction with Puf3 and its ability to boost Cox2 protein

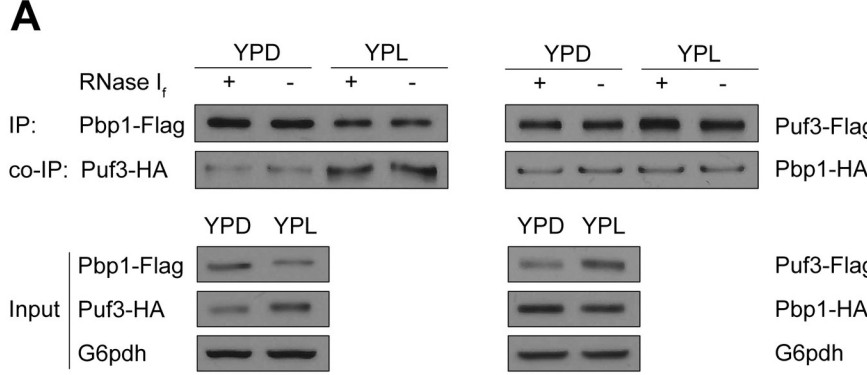

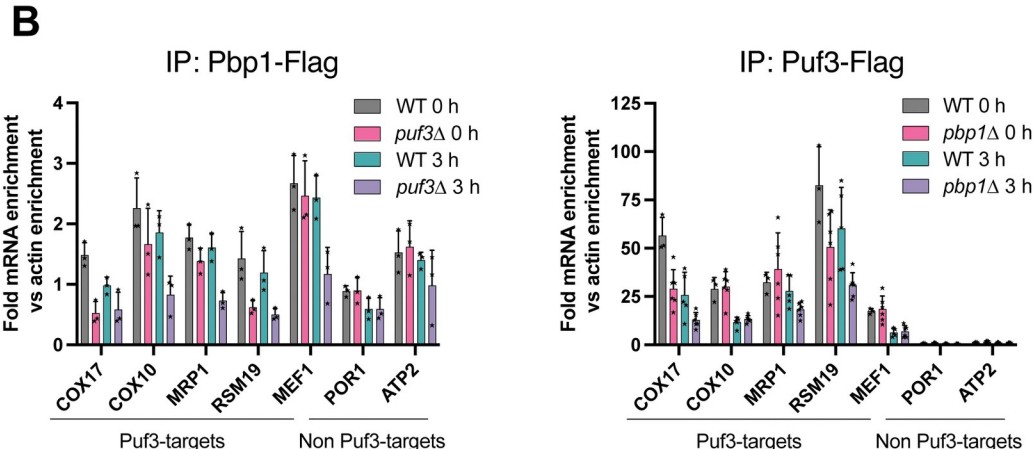

**Fig 7. Pbp1 associates with mRNAs through Puf3.** A. Strains expressing Pbp1-Flag, Puf3-HA or Puf3-Flag, Pbp1-HA were grown in YPD or switched to YPL for 3 h. Flag immunoprecipitation of Pbp1-Flag or Puf3-Flag was performed, and the beads were treated with or without RNase I$_f$ for 15 min to assess whether the associating amount of Puf3-HA or Pbp1-HA is impacted by the presence of any bound RNA. B. RNA IP analysis of transcripts bound to either Pbp1 or Puf3. Strains expressing Pbp1-Flag or Puf3-Flag were grown in YPD or switched to YPL and then associating transcripts were assessed by co-IP followed by qRT-PCR as described in Methods.

amounts. Interestingly, Pbp1 has been suggested to function as a redox sensor [17]. Reactive oxygen species can accumulate during mitochondrial dysfunction, underlining the potential for a key sensor role for Pbp1 in modulating Puf3 functions in response to mitochondrial dysfunction or biogenesis.

Pbp1 has been proposed to regulate mRNA decay and translation through interactions with various RNA regulatory factors including the poly(A) binding protein, the DEAD-box helicase Dhh1/DDX6, and the CCR4-Not and Pan2-Pan3 deadenylase complexes [11,22–25]. This may represent a general mechanism to regulate mRNA decay. However, a recent study reported reduced COX10 and mitochondrial ribosomal protein mRNAs, as well as poor growth of *pbp1Δ* cells in respiratory conditions [26], consistent with our findings that Pbp1 can exhibit specificity for particular transcripts. An increasing body of evidence links Pbp1 and mitochondrial respiratory functions, showing that cells without Pbp1 exhibit increased petite formation and sensitivity to mitochondrial toxins [16,17]. Additionally, Pbp1 overexpression can rescue mitochondrial stress in different settings [13–15]. Our results support the idea that Pbp1 enables and facilitates mitochondrial biogenesis by promoting translation of Puf3-target

mRNAs. Additional interacting partners of Pbp1 and Puf3 are likely involved in stabilizing and promoting the translation of these mRNAs. The specific regulation of transcripts encoding mitochondrial proteins by Pbp1 may explain the various mitochondrial-associated phenotypes observed following loss of Pbp1 function, which may be attributed to the reduced abundance and translation of such transcripts. Although we did not observe any significant defect in the import of Cox4 in *pbp1Δ* cells (S1 Fig), it remains possible that a defect in mitochondrial biogenesis in *pbp1Δ* mutant cells could subsequently result in defects in mitochondrial membrane potential and protein import.

Both Pbp1 and its mammalian ortholog ataxin-2 (ATXN2) harbor conserved domains (Lsm, LsmAD, a methionine-rich LCD), suggesting that both ataxin-2 and Pbp1 may play similar roles in the cell [27,28]. PolyQ expansion mutations in ATXN2 are associated with SCA2 and ALS [29–32]. Murine studies have shown that loss of ataxin-2 alters many transcripts in the adult liver and cerebellum, with a particular increase in mRNAs involved in ribosomal function and translation [33]. At the protein level, loss of ataxin-2 leads to reduced mitochondrial enzymes in the liver and downregulation of fatty acid metabolism pathways, citric acid cycle, and branched-chain amino acids metabolism in both the liver and cerebellum, suggesting that ataxin-2 regulates several metabolic pathways [34]. How ataxin-2 affects different pathways could possibly be explained by the modulation of mRNA levels, such as those involved in the mitochondrial stress response and the mitochondrial quality control factor PTEN-induced kinase 1 (PINK1) [35]. In addition, the C-terminal LCD of *Drosophila melanogaster* ataxin-2 was found to mediate RNP granule formation [36]. Although there is no obvious ortholog of Puf3 in mammalian organisms, PUM1 and PUM2 are mammalian Pumilio proteins whose disruption have been associated with ataxia or mitochondrial phenotypes [37,38]. As such, we speculate that ataxin-2, like Pbp1, might also regulate mRNAs targeted to the mitochondria through interactions with other RNA-binding proteins, perhaps mediated by its C-terminal LCD.

Neurons have a high mitochondrial density and require the maintenance of a functional mitochondrial population throughout cell bodies [39]. The presence of mitochondrial dysfunction in neurons is frequently observed in neurodegenerative diseases, either as a hallmark or a consequence of the disease [40]. Our work uncovering the function of Pbp1 in the regulation of nuclear-encoded mitochondrial gene mRNAs and mitochondrial biogenesis strongly supports the investigation of ataxin-2 as a regulator of mitochondrial functions in the context of neurodegenerative conditions.

## Methods

### Yeast strains, growth, and media

The prototrophic *Saccharomyces cerevisiae* CEN.PK strain [41] was used in all experiments. All strains used in this study are listed in S2 Table. Gene deletions were performed using standard PCR-based strategies to amplify resistance cassettes with appropriate flanking sequences and replace the target gene through homologous recombination [42]. C-terminal tags were similarly made using PCR to amplify resistance cassettes with flanking sequences. Pbp1 and Puf3 mutants with various domain deletions or point mutations were first made using PCR and then integrated into the PBP1 or PUF3 locus in a *pbp1Δ* or *puf3Δ* strain with different selection markers. Yeast strains were grown in YPD (1% yeast extract (Bio Basic), 2% peptone (BD Biosciences) and 2% glucose) or YPL (0.5% yeast extract, 2% peptone and 2% lactate (Sigma L1375)). Cells from overnight cultures were inoculated into fresh YPD to 0.3 optical density ($OD_{600}$)/ml and grown for at least two generations to log phase. Cells were then spun down, washed with YPL, and resuspended in the same volume of YPL. Samples were collected

at indicated time points. For cells treated with rapamycin, 200 ng/ml rapamycin (Sigma) was added to cells grown in YPL and incubated for 30 min before harvesting.

## Growth curves

Cells were inoculated in YPD and grown overnight. The following day, cells were resuspended in fresh media, YPD or YPL, and grown for ~6 h. Cells were then diluted to an $OD_{600}$ of 0.05 in fresh YPD or 0.1 in fresh YPL, and 100 μl of culture was pipetted into a 96 multiwell flat-bottom transparent plate. The plate was shaken and kept at 30˚C while the absorbance was measured every 30 minutes for 18 hours by the SPARK multimode plate reader (TECAN) according to the manufacturer's instructions.

## Whole-cell lysate protein extraction

Cell pellets (5 units of $OD_{600}$) were quenched in 15% TCA for 15 min on ice, pelleted, and washed with ice-cold 100% acetone. Pellets were then resuspended in 240μl urea extraction buffer (6 M urea, 50 mM Tris-HCl pH 6.8, 1 mM PMSF, 1% SDS, 1 mM EDTA, 1 mM DTT, 1 mM $Na_3VO_4\cdot2H_2O$, 5 μM pepstatin A, 10 μM leupeptin, 2X Roche protease inhibitor cocktail), 200 μl glass beads were added, and cells were lysed by bead-beating. After a 5 min incubation at 65˚C, the lysate was centrifuged at maximum speed for 10 min. The supernatant was collected, and the protein concentration was normalized using a BCA protein assay kit (Thermo Scientific). Sample buffer (50 mM Tris-HCl pH 6.8, 2% SDS, 10% glycerol, 10 mM DTT, bromophenol blue) was added.

## Co-immunoprecipitation

At the indicated time points, 50 units of $OD_{600}$ cells were harvested, flash-frozen with liquid nitrogen, and stored at -80˚C until cell lysis. The cell pellet was resuspended in 350 μL of lysis buffer A (50 mM HEPES pH 7.5, 150 mM NaCl, 10 mM $MgCl_2$, 0.5% NP-40, 2X protease inhibitor cocktail (Roche), 1 mM PMSF, 1 mM $Na_3VO_4\cdot2H_2O$, 50 μM pepstatin A, 10 μM leupeptin, 5 mM NaF). After adding 300 μL of glass beads, cells were lysed by bead beating six times: 30 sec of beating / 2 min of cooling on ice. The lysed cells were then separated from glass beads by centrifugation at 6000 rpm for 2 min at 4˚C and diluted with 525 μL of lysis buffer B (buffer A devoid of NP-40). Crude cell extracts were clarified by two successive centrifugations at maximum speed for 10 min at 4˚C. The protein concentration of the cleared lysates was then measured using Bradford assay (Bio-Rad) and adjusted to be equal among all samples in 800 μL reaction volume. For input samples, 10 μL of the reactions was mixed with 30 μl 8 M urea and 12.5 μl 4X SDS sample dye with 40 mM DTT, and denatured for 5 min at 65˚C. For each co-immunoprecipitation reaction, 25 μL of Dynabeads Protein G (Life technologies) was washed with IP-lysis buffer and incubated with 3 μg of mouse anti-Flag M2 antibody (Sigma) for 1 h at 4˚C. Unbound antibody was removed by centrifugation at 500 g for 1 min at 4˚C. The conjugated Dynabeads-antibody were then added to the cleared lysates. After incubating for 1–2 h at 4˚C, the Dynabeads were washed three times with wash buffer (50 mM HEPES pH 7.5, 150 mM NaCl, 1 mM EDTA, 2X protease inhibitor cocktail, 0.2% NP-40). The sample was collected by boiling the beads in 1X SDS sample dye with 6% β-mercaptoethanol for 5 min at 95˚C.

## Western blot

Western blots were carried out by running whole cell lysate or co-IP samples on a NuPAGE 4–12% BisTris gel (ThermoFisher) with MOPS running buffer. Wet transfers at 350 mA at 4˚C

for 75–90 min were used to transfer protein onto a 0.45-micron nitrocellulose membrane. Membranes were blocked with 5% milk in TBST for 1 h and incubated with primary antibody overnight at 4˚C in 1% milk in TBST. Membranes were then washed with TBST and incubated for 1 h with anti-mouse or anti-rabbit HRP antibody in 1% milk in TBST before incubating with Pierce ECL Western blotting substrate and exposing to film. The following antibodies were used: rabbit anti-Flag M2 (Cell Signaling, #2368), rabbit anti-Ha (Cell Signaling, #3724), rabbit anti-G6pdh ab2 (Sigma, #A9521), rabbit anti-Rpn10 (Abcam, #ab98843), mouse anti-Cox2 (ThermoFisher, #459150), mouse anti-Por1 (ThermoFisher, #459500), mouse anti-GFP (Roche, clone 7.1 and 13.1), rabbit anti-Cox4 [43], and rabbit anti-Atp2 [44] (this antibody was a kind gift from Dr. S.M. Claypool at Johns Hopkins University).

## qRT-PCR analysis

Approximately 1 $OD_{600}$ of cells was collected at the indicated time points, flash-frozen, and stored at -80˚C. Total RNA was extracted with MasterPure Yeast RNA Purification Kit following the manufacturer's protocol. cDNA was made with Superscript III reverse transcriptase (Invitrogen), and gene expression levels were analyzed by qRT-PCR with SYBR Green (Invitrogen). mRNA levels were normalized against ACT1. Primers are listed in S3 Table.

## β-estradiol inducible reporter system

Experiments with the Puf3-regulated mRNA reporter were performed as previously described [9]. In brief, cells were grown in YPD to log phase and 100 nM of β-estradiol was added for 45 min to induce transcription. Cells were then spun down and washed twice with YPL to remove residual β-estradiol. Cells were then resuspended in similar volume in YPL and collected at 0, 30, 60, 90, and 180 min to assess the yEGFP mRNA and protein levels.

## RNA-seq analyses

Total RNA was extracted using MasterPure Yeast RNA Purification Kit and samples were sent to Novogene for quality control, poly-A enrichment, library construction, and sequencing. Data quality assessment and read trimming were performed by FastQC (https://www.bioinformatics.babraham.ac.uk/projects/fastqc/) and TrimGalore (https://www.bioinformatics.babraham.ac.uk/projects/trim_galore/), respectively. Paired end reads with Phred scores greater than 35 were mapped to the yeast reference genome (S288C) using STAR version 2.7.3a [45] and the alignment statistics were analyzed by Samtools flagstat [46]. Uniquely mapped reads with minimum mapping quality scores of 5 were quantified per gene ID using featureCounts [47] from Subread package (v1.6.3) and reference annotations (R64-1-1). R package DESeq2 (v 3.16) [48] was used to identify differentially expressed genes. A gene is considered significantly differentially expressed with cutoff values of false discovery rate (FDR) less than 0.05 and absolute $log_2$fold change greater than 0.58. Gene ontology (GO) terms that were significantly enriched in sets of differentially expressed genes were obtained using the R package clusterProfiler (v 4.2) [48]. All statistical analyses were performed in RStudio v 2022.07.2+576. Raw sequencing data have been deposited at Gene Expression Omnibus with accession number: GSE227356.

## Transcriptional shut-off assay

Cells were grown in YPD to log phase then switched to YPL without back-dilution and cultured for 3 h. Thiolutin was added at a final concentration of 3 µg/mL and 1 mL samples were collected at the indicated time-points, flash-frozen, and stored at -80˚C. RNA extraction and

qRT-PCR analysis were performed as described above. PGK1 was used as an internal control gene due to its relative mRNA stability following thiolutin treatment [49,50].

## RNA IP

Approximately 50 ODs Pbp1-FLAG cells were collected in YPD and after 3 h in YPL by centrifugation and freezing in liquid nitrogen. The frozen pellets were lysed in 0.4 mL lysis buffer (100 mM Tris-Cl pH 7.5, 50 mM NaCl, 0.2% Tween-20, 10% glycerol, 0.1% β-mercaptoethanol, 1X EDTA-free protease inhibitor cocktail (Roche), 1 mM EDTA, 1 mM EGTA, 1 mM PMSF, 5 μM Pepstatin A, 10 μM Leupeptin, 0.2 mM sodium orthovanadate, 10 mM β-glycerophosphate, 60 mM NaF, 10 mM $NaN_3$ and 10U Riboguard (Biosearch Technologies)). Cells were lysed by bead beating and resulting lysates were cleared by centrifugation at 21,100g for 10 min. An 8 μl aliquot was removed for input RNA extraction and the remaining lysate was incubated with 25 μl Protein G Dynabeads (LifeTech) preincubated with 5 μl anti-FLAG M2 antibody (Sigma) at 4˚C for 1 h. Beads were then washed 3X with 1 mL lysis buffer. 1/8 of the beads were treated with SDS sample buffer to elute bound Pbp1-FLAG and subjected to SDS-PAGE and Western blot for protein quantification. RNA was purified from the remaining 7/8 beads, as well as the input lysate aliquot, using the MasterPure Yeast RNA Purification Kit (Biosearch Tech). qRT-PCR was performed using SYBR Green (Invitrogen). The target recovered was calculated by dividing the co-IP target RNA abundance by the input target RNA abundance. This was then divided by recovered actin to give relative enrichment.

## Mitochondrial precursor protein analysis

Yeast cells were grown in YPL at 30˚C up to logarithmic phase. Cultures were then treated with 0.1% DMSO or 40 μM CCCP (carbonyl cyanide *m*-chlorophenyl hydrazone) for 6 h. For each sample, 5 $OD_{600}$ was collected and analyzed by SDS-PAGE and immunoblotting.

## Supporting information

**S1 Fig. Unprocessed Cox4 does not accumulate in *pbp1Δ* cells upon membrane potential dissipation.** (A) Total protein cell extracts from Pbp1-FLAG or *pbp1Δ* yeast grown in YPL at 30˚C and supplemented with 0.1% DMSO or 40 μM CCCP for 6 h. Pbp1-FLAG, *cox4Δ* strain serves as control for antibody specificity. p:precursor, m:mature.
(TIF)

**S2 Fig. Effect of thiolutin treatment on Puf3 target mRNAs in WT and *pbp1Δ* mutant cells.** Puf3 target mRNAs in *pbp1Δ* cells are relatively resistant to decay following treatment with thiolutin. Cells growing at log-phase in YPL media were treated with 3 μg/mL thiolutin and collected at the indicated time-points. The plots represent the fold change in mRNA levels with respect to t = 0 calculated by RT-PCR with normalization to PGK1 mRNA. Error bars represent SD; n = 4.
(TIF)

**S3 Fig. Stability of Pbp1p and Puf3p in the absence of one another.** (A) Puf3 protein levels are decreased in cells lacking Pbp1. Cells expressing Puf3-Flag were assayed in the presence or absence of Pbp1 for the indicated proteins by Western blot. (B) Pbp1 protein levels are not affected in cells lacking Puf3. Cells expressing Pbp1-Flag were assayed in the presence or absence of Puf3 for the indicated proteins by Western blot at various time points following switch from YPD to YPL.
(TIF)

**S4 Fig. Transcript abundance in *pbp1Δ*, *puf3Δ*, and *pbp1Δpuf3Δ* mutant cells.** Quantitation of the same panel of transcripts as in Fig 1C in the indicated strains.
(TIF)

**S5 Fig. Transcript abundance in various Pbp1 or Puf3 mutant strains.** Several mRNAs involved in mitochondrial function were assayed by qRT-PCR in the indicated Pbp1 (A) or Puf3 (B) mutant strains at the indicated time points after switch from YPD to YPL. The abundance of the indicated transcripts was normalized to ACT1.
(TIF)

**S6 Fig. Decreased Cox2 levels in *pbp1Δ* cells are not merely due to hyperactive TORC1 signaling.** The indicated strains were grown in YPL for 3 h, of which the last 30 min were in the absence (-) or presence (+) of 200 ng/mL rapamycin. Samples were collected, quenched, and extracted for immunoblot analysis of the indicated proteins. Note that rapamycin treatment did not restore protein amounts of Cox2 in *pbp1Δ* cells. Npr1-HA, a TORC1-dependent substrate, was used as a positive control for rapamycin.
(TIF)

**S1 Table. RNA-seq expression analysis of *pbp1Δ* mutant cells.** Spreadsheet containing raw read counts of transcripts expressed in *pbp1Δ* and WT cells grown in YPD (glycolytic) or switched to YPL (respiratory) for 3 h. These expression data were used for the analyses depicted in Fig 2.
(XLSX)

**S2 Table. Saccharomyces cerevisiae strains used in this study.**
(PDF)

**S3 Table. qRT-PCR primers used in this study.**
(PDF)

## Acknowledgments

We thank the Tu Lab for helpful discussions. We thank Dr. S.M. Claypool at Johns Hopkins University for the Atp2 and Cox4 antibodies.

## Author Contributions

**Conceptualization:** Floortje van de Poll, Yu-San Yang, Chien-Der Lee, Benjamin P. Tu.

**Data curation:** Samira Jahangiri.

**Formal analysis:** Floortje van de Poll, Benjamin M. Sutter, Michelle Grace Acoba, Daniel Caballero, Samira Jahangiri.

**Investigation:** Floortje van de Poll, Benjamin M. Sutter, Michelle Grace Acoba, Daniel Caballero, Samira Jahangiri.

**Methodology:** Floortje van de Poll, Benjamin M. Sutter, Michelle Grace Acoba, Daniel Caballero, Samira Jahangiri.

**Project administration:** Benjamin P. Tu.

**Supervision:** Benjamin P. Tu.

**Writing – original draft:** Floortje van de Poll.

**Writing – review & editing:** Floortje van de Poll, Benjamin M. Sutter, Michelle Grace Acoba, Daniel Caballero, Benjamin P. Tu.

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
