## [Decision Letter · Decision Letter 0]

7 May 2023

Dear Dr. Tu,

Your manuscript entitled "Pbp1 associates with Puf3 and promotes translation of its target mRNAs involved in mitochondrial biogenesis" has been reviewed by two of the experts who reviewed the original work submitted to Research Commons; it has also be editorially reviewed. We are pleased to inform you that all the evaluators of your current version agree that your studies will be an important contribution and therefore we accept this interesting study for publication in PLOS Genetics. Congratulations!

Yours sincerely,

Anita K. Hopper

Academic Editor

PLOS Genetics

Gregory Barsh

Editor-in-Chief

PLOS Genetics

Comments from the reviewers (if applicable):

Reviewer's Responses to Questions

**Comments to the Authors:**

Reviewer #1: The authors have adequately responded former reviewer's comments. So I think that this article is suitable for publication in PLOS Genetics.

Reviewer #2: The authors have done a commendable job in addressing prior concerns. I have no further critiques and recommend acceptance.

**Have all data underlying the figures and results presented in the manuscript been provided?**

Reviewer #1: Yes

Reviewer #2: Yes

PLOS authors have the option to publish the peer review history of their article (what does this mean?). If published, this will include your full peer review and any attached files.

Reviewer #1: No

Reviewer #2: No

**Data Deposition**

http://datadryad.org/submit?journalID=pgenetics&manu=PGENETICS-D-23-00380

**Press Queries**

---

## [Editor Report · Acceptance letter]

18 May 2023

PGENETICS-D-23-00380 

Pbp1 associates with Puf3 and promotes translation of its target mRNAs involved in mitochondrial biogenesis 

Dear Dr Tu, 

We are pleased to inform you that your manuscript entitled "Pbp1 associates with Puf3 and promotes translation of its target mRNAs involved in mitochondrial biogenesis" has been formally accepted for publication in PLOS Genetics! Your manuscript is now with our production department and you will be notified of the publication date in due course.

With kind regards,

Zsofia Freund

PLOS Genetics

On behalf of:
